# The Placental NLRP3 Inflammasome and Its Downstream Targets, Caspase-1 and Interleukin-6, Are Increased in Human Fetal Growth Restriction: Implications for Aberrant Inflammation-Induced Trophoblast Dysfunction

**DOI:** 10.3390/cells11091413

**Published:** 2022-04-21

**Authors:** Irvan Alfian, Amlan Chakraborty, Hannah E. J. Yong, Sheetal Saini, Ricky W. K. Lau, Bill Kalionis, Evdokia Dimitriadis, Nadia Alfaidy, Sharon D. Ricardo, Chrishan S. Samuel, Padma Murthi

**Affiliations:** 1Department of Pharmacology, Monash Biomedicine Discovery Institute, Monash University, Melbourne, VIC 3800, Australia; irvanalfian@gmail.com (I.A.); amlan.chakraborty@monash.edu (A.C.); sheetal.saini@monash.edu (S.S.); wai.lau1@monash.edu (R.W.K.L.); sharon.ricardo@monash.edu (S.D.R.); 2Faculty of Medicine, Universitas Indonesia, Jl. Salemba Raya 6, Jakarta Pusat 10160, Indonesia; 3Singapore Institute for Clinical Sciences, Agency for Science, Technology and Research, Singapore 117609, Singapore; hannah_yong@sics.a-star.edu.sg; 4Department of Maternal-Fetal Medicine Pregnancy Research Centre, The Royal Women’s Hospital, Melbourne, VIC 3052, Australia; bill.kalionis@thewomens.org.au; 5Department of Obstetrics and Gynaecology, The University of Melbourne, Melbourne, VIC 3052, Australia; eva.dimitriadis@unimelb.edu.au; 6Gynaecology Research Centre, The Royal Women’s Hospital, Melbourne, VIC 3052, Australia; 7Institut National de la Santé et de la Recherche Médicale U1292, Biologie et Biotechnologie pour la Santé, 38043 Grenoble, France; nadia.alfaidy-benharouga@cea.fr; 8Commissariat à l’Energie Atomique et aux Energies Alternatives (CEA), Biosciences and Biotechnology Institute of Grenoble, 38054 Grenoble, France; 9Service Obstétrique & Gynécologie, Centre Hospitalo-Universitaire Grenoble Alpes, University Grenoble-Alpes, CEDEX 9, 38043 Grenoble, France

**Keywords:** fetal growth restriction, placental function, inflammasomes, NLRP3, caspase-1, cytokines, apoptosis

## Abstract

Fetal growth restriction (FGR) is commonly associated with placental insufficiency and inflammation. Nonetheless, the role played by inflammasomes in the pathogenesis of FGR is poorly understood. We hypothesised that placental inflammasomes are differentially expressed and contribute to the aberrant trophoblast function. Inflammasome gene expression profiles were characterised by real-time PCR on human placental tissues collected from third trimester FGR and gestation-matched control pregnancies (n = 25/group). The functional significance of a candidate inflammasome was then investigated using lipopolysaccharide (LPS)-induced models of inflammation in human trophoblast organoids, BeWo cells in vitro, and a murine model of FGR in vivo. Placental mRNA expression of *NLRP3*, caspases 1, 3, and 8, and interleukin 6 increased (>2-fold), while that of the anti-inflammatory cytokine, *IL-10*, decreased (<2-fold) in FGR compared with control pregnancies. LPS treatment increased NLRP3 and caspase-1 expression (>2-fold) in trophoblast organoids and BeWo cell cultures in vitro, and in the spongiotrophoblast and labyrinth in the murine model of FGR. However, the LPS-induced rise in NLRP3 was attenuated by its siRNA-induced down-regulation in BeWo cell cultures, which correlated with reduced activity of the apoptotic markers, caspase-3 and 8, compared to the control siRNA-treated cells. Our findings support the role of the NLRP3 inflammasome in the inflammation-induced aberrant trophoblast function, which may contribute to FGR.

## 1. Introduction

Fetal growth restriction (FGR) is defined as the failure of the fetus to reach its full growth potential in utero and is characterised by a deceleration or stagnation in the fetal growth trajectory, placing the fetus at high risk of stillbirth or iatrogenic preterm birth [1]. As well as the immediate consequences of being born too small, leading to increased perinatal morbidity and mortality, affected offspring are at increased risk of metabolic syndrome including pulmonary and cardiovascular disorders, diabetes, and other chronic diseases in later life [2]. The prediction, detection, and treatment of pregnancies at a high risk of FGR remain a major challenge, with early delivery the only option to prevent stillbirth [3]. This is largely due to a lack of detailed understanding of the molecular aetiology of FGR, and the crucial role that placental dysfunction plays in this disorder [4].

The aetiology of FGR is complex and includes maternal, fetal, and placental abnormalities [5]. These identifiable causes of FGR account for about 40% of cases, with the remainder being idiopathic [6]. Idiopathic FGR is commonly associated with placental insufficiency [5] and is characterised by evidence of fetal health compromised by oligohydramnios and asymmetric fetal growth, featuring an increased head to abdominal circumference ratio. Evidence of an underlying pathology associated with FGR allows clinicians to distinguish between FGR and healthy constitutionally small for gestational age (SGA) babies that are otherwise normal.

Typically, a FGR-affected placenta is smaller than its gestation-matched control, with associated placental villous malformations including a decrease in the number of villi, villous diameter, and surface area, as well as a decrease in arterial number, lumen size, and villous branching [7]. At the cellular level, the FGR-affected placenta is characterised by increased premature differentiation of the villous cytotrophoblasts and apoptosis [8,9]. Additionally, there is reduced nutrient transporter activity, reduced extravillous cytotrophoblast (EVCT) proliferation, migration leading to a shallow invasion of the EVCT into the maternal decidua, and consequently, an abnormal remodelling of the spiral arterioles [10]. The causes of placental abnormalities detected in FGR are unknown, but there is evidence of placental inflammation, hypoxia, and subsequent oxidative and/or nitrative stress. These abnormalities are attributed to placental dysfunction and aberrant placental perfusion in pregnancies complicated by FGR [4,11,12]. However, the mechanistic pathways underlying placental dysfunction in FGR have not been fully elucidated.

This study aimed to identify markers specifically targeting placental inflammation-induced pathogenesis of FGR. The role of the inflammasome machinery in contributing to placental inflammation in FGR has been poorly studied. Inflammasomes are cytosolic, self-assembling, multimeric protein complexes that organise host defence mechanisms in response to pathogens, and they also regulate cytokine production. Inflammasomes are activated either by the presence of pathogen-associated molecular patterns (PAMPs) or damage-associated molecular patterns (DAMPs) [13]. PAMPs are released during a bacterial infection or expressed by microbes, whereas sterile inflammation, caused by damaged host cells, leads to the expression of DAMPs in multiple forms including uric acid crystals, ATP, exosomes, cholesterol, and heat-shock proteins [14,15]. PAMP and DAMP activation is detected by pattern recognition receptors (PRRs), such as toll-like receptors (TLRs), NOD-like receptors (NLRs), RIG-I-like receptors (RLRs), C-type lectin receptors (CLRs), and AIM2-like receptors (ALRs) [14]. NLRs and TLRs are the two major families of PRRs that regulate the expression and production of pro- and anti-inflammatory cytokines [16]. Recently, we provided evidence of a significant function for NLRP7 in human FGR [17] and described the role of NLRP7 in the development of choriocarcinoma [18,19]. In this study, we hypothesised that placental inflammasomes are differentially expressed and contribute to aberrant cytokine expression in FGR pregnancies. This current study characterised the expression profile of inflammasomes and their downstream pro-and anti-inflammatory cytokines in the placentae obtained from FGR and gestation-matched uncomplicated normal (control) pregnancies. We also investigated how candidate inflammasomes modulate placental cytokines, and assessed differentiation marker expression and apoptosis using two independent in vitro models of trophoblasts, and in the placentae obtained from an in vivo inflammation-induced FGR in a murine model of pregnancy.

## 2. Materials and Methods

### 2.1. Human Placental Tissues

Human placental tissue samples from first trimester termination of pregnancies and from third trimester FGR and gestation-matched control pregnancies were obtained with written consent and with the approval of The Royal Women’s Hospital Human Research and Ethics Committees (RWH-HREC), Melbourne, Australia. Placentae from FGR pregnancies were collected, with a gestational age ranging from 27–41 weeks, while uncomplicated control samples were matched for gestational age. As specified by the Australian fetal growth charts, FGR subjects included in this study had an infant birth weight of <10th percentile, with the addition of at least one of the other ultrasound criteria: abnormal umbilical artery Doppler velocimetry (elevated, reversed, or absent); asymmetric growth (head circumference: abdominal circumference ratio, HC:AC > 1.2); and an abnormal amniotic fluid index (AFI ≠ 7). There were no clinical complications in any control patients, who delivered normal and healthy babies with appropriate birth weight for gestation. Control samples were further classified into pre-term and term controls. All pre-term control patients underwent spontaneous pre-term vaginal delivery, or required elective delivery with non-obstetric complications, and had no association with placental insufficiency. The exclusion criteria for both the control and FGR samples included in this study were preeclampsia, prolonged rupture of the membranes beyond 24 h, maternal chemical dependency, multiple pregnancies, placental abruption, fetal congenital anomalies, and suspicion of intrauterine viral infection [20,21,22]. Placentae were collected within 20 min of the delivery of the placenta, and the villous tissue was dissected away from the decidua, washed in phosphate-buffered 0.9% (PBS) solution to prevent blood contamination, and snap frozen or fixed in formalin/cryomatrix and stored at −80 °C until use [20,21,22].

### 2.2. Fluidigm Biomark^TM^ Array

To screen for the expression of inflammasome genes in the placental tissues obtained from FGR and gestation-matched control pregnancies, a high-throughput real-time PCR system called Fluidigm Biomark^TM^ Array was performed at the Monash Health Genomic Facility (Clayton, Melbourne, Australia), according to the manufacturer’s instructions and as previously described [23]. Briefly, total placental RNA was extracted from approximately 500 mg of placental tissues (collected and pooled from different areas of each individual placenta) using a Qiagen Midi RNeasy kit (Qiagen, Hilden, Germany). Two µg of RNA from the cases and control samples was reverse transcribed using the First Strand Synthesis kit and Superscript III (Invitrogen, Waltham, MA, USA). Then, approximately 12.5 ng/μL/well of cDNA was added to the Fluidigm Biomark^TM^ Array. Two independent housekeeping genes, *18S rRNA* and *GAPDH*, were used in this study as endogenous reference genes for relative gene expression quantification [24]. The array plate included primer sets for key components genes of the inflammasomes including *NLRP3, TLR2, TLR5, TLR6, NLRC5, Caspase 1 (CASP1), interleukin-1 β (IL-1β), nuclear factor kB (NFκB),* and *NOD2*. The array also included primer sets for pro- and anti-inflammatory cytokine genes and marker genes of apoptosis, which were previously identified as downstream targets of inflammasome signalling [25,26] such as *Caspase 3 (CASP3), Caspase 8 (CASP8), interferon γ (IFNγ), interleukin-6 (IL-6),* and *interleukin-10 (IL-10)*.

### 2.3. Independent Validation by Using Real-Time PCR

To identify any pathological expression differences of inflammasome genes, with detectable expression as shown by the Fluidigm Biomark^TM^ array, real-time PCR was performed using inventoried TaqMan probes (Thermo Fisher Scientific, Waltham, MA, USA). The genes tested were *NLRP3, NLRC5, CASP1, CASP3, CASP8, NOD2, NFκB1, IFNγ, IL-1β, IL-6,* and *IL-10* on the 7700 real-time PCR instrument (Applied Biosystems, Thermo Fisher Scientific, Waltham, MA, USA). Semi-quantitation of the genes of interest relative to the endogenous control gene (*18S rRNA)* was determined using the 2^−ΔΔCT^ method as previously described [27].

### 2.4. 3D Cultures of Trophoblast Organoids In Vitro

The functional significance of the candidate inflammasomes, which showed an increased expression in the placentae obtained from human FGR pregnancies, was further investigated using a low-dose lipopolysaccharide (LPS, 1 ng/mL)-induced in vitro model of inflammation in human first-trimester placental trophoblast organoid cultures. A low-dose LPS was chosen to mimic sub-clinical infection, and the dosage was decided based on previously well characterised data showing that this low-dose was physiologically relevant, maintained placental tissue integrity, and phenotypic and functional changes were induced as previously observed in vivo in FGR placentas [12]. Briefly, placental trophoblast organoids were prepared from human chorionic villous tissues obtained after 6–12 weeks of gestation following the termination of pregnancies at the Royal Women’s Hospital. Samples were collected with written consent from the patients and with the approval of the RWH-HREC, Melbourne. Organoids were prepared from mononuclear cytotrophoblasts, with modifications of the protocol previously described by Haider et al. [28]. Briefly, villous cytotrophoblasts (VCT) were isolated using an enzymatic mixture containing trypsin (0.25%) and DNase I (1.5 mg/mL) in phosphate buffered saline (PBS). Trophoblast organoids were prepared by culturing isolated mononuclear VCTs in Cultrex^®^ Matrigel Reduced Growth Factor Basement Membrane Extract (BME), and Type R1 (Corning, NY, USA) in a trophoblast organoid medium (TOM) supplemented with Advanced DMEM/F12, B27, and N2 Supplements, L-glutamine, 10 mM HEPES, 0.5% Penicillin/Streptomycin (Gibco, Waltham, MA, USA), 3 μM CHIR99021(Sigma-Aldrich, St. Louis, MO, USA), 1 μM A83-01 (Sigma-Aldrich, St. Louis, MO, USA), and 100 ng/mL EGF (Abcam, Cambridge, UK). Cultures were maintained in 5% CO_2_/95% air in a humidified incubator at 37 °C and the medium was replaced every 2–3 days. Small organoid clusters became visible around day 7, and the organoid cultures were further sub-cultured to passage 3. The effect of inflammation on the growth of trophoblast organoids, and the concentrations of NLRP3 and caspase-1 in the culture media of the organoid clusters, were determined by treating them with LPS (1 ng/mL in growth factor reduced TOM) for 72 h. The trophoblast organoid growth over 72 h, following treatment with LPS, was determined by immunostaining using the proliferating marker, Ki67; and the NLRP3 protein was localised using immunofluorescence, as previously described by Rezanejad et al. [29].

### 2.5. Trophoblast-Derived Cell-Line, BeWo In Vitro

The functional consequences of an increased NLRP3 expression were investigated in an in vitro model of LPS-induced inflammation involving a monolayer culture of a trophoblast-derived cell line, BeWo. The choriocarcinoma-derived BeWo cell line, which served as a model for villous cytotrophoblasts (gift from Prof Stephen Rogerson, Department of Medicine, Royal Melbourne Hospital and University of Melbourne, Melbourne, Australia), was maintained in a medium of RPMI-1640 supplemented with 10 mM sodium bicarbonate, 100μg/mL streptomycin, 50 IU/mL penicillin, and 10% fetal bovine serum (FBS). Confluent cultures of BeWo were serum starved overnight in RPMI-1640 medium supplemented with 1% bovine serum albumin. The cells were then induced to aggregate and fuse with 100 µM forskolin used to promote differentiation into syncytiotrophoblast (STB). To assess the effect of inflammation, BeWo cells treated with forskolin were incubated with low-dose LPS (1 ng/mL) to mimic sub-clinical infection for 72 h [12].

In a separate set of experiments, LPS-treated BeWo cells were transfected with NLRP3-specific siRNA (siNLRP3), or negative control (siCONT), and similarly treated with LPS to determine the role of NLRP3 in the response to LPS. Briefly, BeWo (2 × 10^5^ cells/well in 6-well plates, and 5 × 10^4^ cells/well in 24-well plates) were transfected with siRNA specific for *NLRP3* (*siNLRP3*, Thermo Fisher Scientific, Waltham, MA, USA). The negative control siRNA (*siCONT*) (AllStars Neg. siRNA AF 488, Qiagen, Hilden, Germany) consisted of a pool of enzyme-generated siRNA oligonucleotides of 15–19 base pairs that were not specific for any known human gene and showed no sequence similarity to *NLRP3*. Briefly, *siNLRP3* or *siCONT* prepared at a final concentration of 80 µM siRNA in a ratio of 1:6 with the Hi-Perfect transfection reagent (Qiagen, Hilden, Germany) was allowed to complex for 15 min at room temperature, then added dropwise to the wells, and incubated for further 72 h in culture. At the end of the incubation periods, NLRP3; caspase-1; cytokines, IL-6, IL-18, IL-1β, and IL-10; chorionic gonadotropin CGB/βhCG, a marker of trophoblast differentiation; and the activity of caspases 3 and 8 were determined in the cellular extracts and in the culture media using real-time PCR and immunoassays, respectively [30,31].

### 2.6. Placental NLRP3 Expression in an Inflammation-Induced Murine Model of FGR, In Vivo

The NLRP3 gene is well conserved between humans and mice and is denoted as NLRP3 [32,33]. The placental expression of the NLRP3 protein in an inflammation-induced murine model of FGR is largely unknown. Therefore, in this study, the spatial distribution of candidate inflammasome proteins in the placentae obtained from Balb/c mice at E18, following LPS-induced inflammation, was investigated. A model of LPS with subclinical infection was included as placental inflammation and elevated DAMPs frequently detected, despite no clinical signs of infection [34,35]. Briefly, pregnant Balb/c mice housed under conventional conditions, had ad libitium access to food and water, and were maintained in a 12 h light–12 h dark cycle. All procedures were approved by Monash University Animal Ethics Committee and followed the NHMRC Australian code of practice for the care and use of animals for scientific purposes. Inflammation in the pregnant Balb/c female mice (n = 4–6) was induced at the mid-gestational period (gestational day (GD) 14), by an intraperitoneal injection of low-dose LPS (1 μg/kg body weight) in 0.1 mL sterile 0.9% saline (*w*/*v*). Control animals (n = 4–6) matched for gestation received saline only. Mice were killed on E18 by carbon dioxide inhalation followed by cardiac puncture to collect peripheral blood. Implantation sites (at least 5/mouse) were dissected to obtain placentae and fetuses. Placentae and fetuses were weighed. Placentae were fixed in 10% neutral buffered formalin. The effect of inflammation on placental NLRP3 and caspase-1 protein localisation was investigated using immunohistochemistry [36].

### 2.7. Immunohistochemistry

Paraffin-embedded 5 µm placental tissue sections obtained from the murine model of pregnancy were deparaffinised in xylene, and then dehydrated in graded alcohol as previously described [36]. Briefly, tissue sections were then incubated overnight with primary rabbit monoclonal anti-NLRP3 (D4D8T, #15101, (1:200) Cell Signaling Technology, Danvers, MA, USA) or caspase-1 (#3866, 1:500 Cell Signaling Technology, Danvers, MA, USA), in 2% (*w*/*v*) bovine serum albumin (BSA) in PBS. Control sections were incubated with 0.02 µg/µL rabbit IgG, 2% (*w*/*v*) BSA in PBS (Dako, Copenhagen, Denmark). Staining was visualised by incubating with the biotinylated secondary antibody and streptavidin-conjugated enzyme. Chromogenic detection was performed using DAB (Sigma Aldrich, St. Louis, MO, USA). Sections were mounted with DPX.

### 2.8. Immunofluorescence

NLRP3 protein localisation in the trophoblast organoids, first trimester and third trimester FGR, and uncomplicated term placental tissues (n = 6) was determined using immunofluorescence, as previously described [29,36]. Briefly, whole mounts of trophoblast organoids grown in 8-well chamber slides were fixed in 5% paraformaldehyde and blocked with 1% BSA/PBS. Tissue sections were dewaxed and rehydrated in graded ethanol. Both organoids and tissue sections were then incubated overnight with rabbit polyclonal anti-NLRP3 or caspase-1 (Cell Signalling Technology, Danvers, MA, USA), as described above. Control slides included sections that were incubated with rabbit IgG in 1% BSA/PBS (negative control), or with the proliferating marker Ki67, using the rabbit polyclonal Ki67 (#15580, 1:200, Abcam, Cambridge, UK). Fluorescence detection was performed using Alexa Fluor 488 or 555 (1:1000, Thermo Fisher Scientific, Waltham, MA, USA) and counter-stained with the nuclear counterstain, 6-diamidino-2-phenylindole, dihydrochloride (DAPI, Dako, Copenhagen, Denmark), according to the manufacturers’ recommendations.

### 2.9. Western Immunoblotting

Total protein was extracted from FGR (n = 5) and gestation-matched controls (n = 5). Immunoblotting for NLRP3 was performed as previously described [36]. Rabbit monoclonal anti-human NLRP3 (D4D8T, #15101, 0.1 µg/mL, 1:1000 dilution; Cell Signaling Technology, Danvers, MA, USA) was used as the primary antibody, and the immunoreactive protein was visualised using a peroxidase-conjugated donkey anti-rabbit IgG-HRP secondary antibody (Zymed, Waltham, MA, USA), and detected by an enhanced chemiluminescence system (Bio-Rad, Hercules, CA, USA). The level of immunoreactive proteins was determined semi-quantitatively using scanning densitometry (Image Quant, Molecular Dynamics Inc., Sunnyvale, CA, USA), relative to rabbit polyclonal GAPDH (1 μg/mL, 1:1000 dilution, Imgenex Technology Corporation, British Columbia, Canada) [36].

### 2.10. Immunoassays

The concentration of NLRP3 was measured using an indirect enzyme linked immunosorbent assay, ELISA (D4D8T, #15101, 1:1000 dilution; Cell Signaling Technology, Danvers, MA, USA), while caspase-1 concentration was measured using either a Quantikine ELISA kit (#DCA 100, R&D Systems, Minneapolis, MN, United States) or the indirect ELISA ((#3866, 1:1000 dilution; Cell Signaling Technology, Davers, MA, USA) in the conditioned media collected from cultured BeWo and trophoblast organoids, according to the manufacturer’s protocol. For the determination of β-hCG protein levels, ELISA (Alpha Diagnostic International, San Antonio, TX, USA) was performed following the manufacturer’s instructions. As previously described, the minimum concentration of human HCG detected using this assay was 1.5 mlU/mL [36]. Assessments of IL-6, IL-18, IL-1β, and IL-10 concentrations were performed using an ELISA, according to the manufacturer’s instructions (Life Technologies, Carlsbad, CA, USA) [11,28]. All samples were assayed in duplicates.

### 2.11. Caspase 3 and Caspase 8 Activity Assays

Caspases 3 and 8 activities were measured in BeWo cells using the ApoAlert^®^ Caspase Colorimetric assay kit specific for the caspase 3/8 activity, according to the manufacturer’s instructions (Clontech Laboratories Inc., Mountain View, CA, USA). Chromogen absorbance was measured at 405 nm using the SPECTRAmax PLUS microplate reader (Molecular Devices Corp., Sunnyvale, CA, USA) [31].

### 2.12. Data Analysis

Differences in the mRNA and protein expression of the components of the placental inflammasomes were analysed using GraphPad Prism V8.0. (GraphPad Software Inc, San Diego, CA, USA). Data analysis for the patient demography and clinical characteristics were analysed as continuous variables for the gestational age, maternal age, placental weight, and birth weight using an unpaired *t*-test and presented as mean ± standard deviation (SD). Fisher’s exact test was performed on two categorical patient characteristics including parity (primiparous and multiparous) and infant sex (male and female). The modes of delivery consisted of three categorical variables (vaginal delivery, caesarean in labour, and caesarean not in labour), and were analysed using a Chi-square test. Gene and protein expression differences were analysed using the Mann–Whitney U Test and Student’s *t*-test, respectively. *p* < 0.05 was considered significant and denoted by *.

## 3. Results

Placental expression of inflammasomes was investigated in FGR and gestation-matched, uncomplicated control pregnancies.

### 3.1. Placental Inflammasomes in FGR

#### 3.1.1. Patient Demography of the Samples 

Table 1 details the patient demography of the third trimester FGR and uncomplicated pregnancies included in this study. As shown, a significant decrease in placental and birth weight was observed in FGR compared with control pregnancies. However, there was no significant difference in maternal age, gestation, or parity between the two groups.

#### 3.1.2. Clinical Criteria of FGR Samples

As shown in Table 2, all FGR pregnancies included in this study are clinically characterised based on birth weight <10th percentile (BW < 10th percentile) and three other ultrasound-determined selection criteria: abnormal umbilical artery Doppler velocimetry, asymmetric growth, and abnormal amniotic fluid index (AFI). All FGR samples fulfill the first criterion for the BW <10th centile and at least one of the other ultrasound criteria. Furthermore, 14 out of 25 samples met all three ultrasound criteria, 9 out of 25 samples met only two ultrasound criteria, and 2 out of 25 samples met only one ultrasound criteria.

HC, head circumference; AC, abdominal circumference.

#### 3.1.3. Screening for the Presence or Absence of Inflammasomes by Fluidigm Biomark^TM^ Array

A Fluidigm Biomark^TM^ Array was performed to screen for the presence or absence of 14 genes of interest: *NLRP3, CASP1, CASP3, CASP8, NFκB1, NLRC5, NOD2, TLR2, TLR5, TLR6, IL-6, IL-10, IL-1β*, and *IFNγ* in the placentae obtained from FGR and control pregnancies at the Monash Genomic Facility (Monash Health Translational Precinct (MHTP), Monash Health. Out of the 14 genes, 7 genes were involved in the DAMPs pathway (*NLRP3, CASP1, NFκB1, CASP3, CASP8, NLRC5,* and *NOD2*), 3 genes were included in the PAMPs pathway (*TLR2, TLR5,* and *TLR6*), and 4 genes were cytokines (*IL-6, IL-10, IL-1β*, and *IFNγ*). Two endogenous control genes, *18S rRNA* and *GAPDH,* were used as house-keeping genes as per the manufacturer’s instructions. All genes for DAMPs (Table 3), PAMPs (Table 4), and cytokines (Table 5) were expressed in all samples from both FGR and control pregnancies.

Values of all mRNA gene expressions relative to the combination of the two endogenous control genes, *18S rRNA* and *GAPDH*, are presented as median ± IQR. IQR refers to inter-quartile range, a measure of statistical dispersion/ spread of the data.

Values of all mRNA gene expressions relative to the combination of the two endogenous control genes, 18S rRNA and GAPDH, are presented as median ± IQR.

Values of all mRNA gene expressions relative to the combination of the two endogenous control genes, *18S rRNA* and *GAPDH*, are presented as median ± IQR.

### 3.2. Independent Validation by Real-Time PCR

In the Fluidigm Biomark^TM^ array, the cycle threshold for the endogenous control gene GAPDH was highly variable, while that of 18S rRNA was consistent (data not shown). Therefore, we only used 18S rRNA as the housekeeping gens for independent validation using real-time PCR with pre-validated probes. As shown in Figure 1 and Figure 2, seven DAMPs-involved gene expression profiles (*NLRP3*, *CASP1*, *NFκB1*, *CASP3*, *CASP8*, *NLRC5*, and *NOD2*) and four cytokine gene expression profiles (*IL-1β*, *IL-6*, *IFNγ*, and *IL-10*) are analysed relative to the endogenous control gene, 18S rRNA, and the gene expression is compared between the FGR and control groups, respectively. All genes tested show significant increases in placental tissues obtained from FGR compared with gestation-matched control pregnancies (*p* < 0.001). In these validated assays, IL-10 mRNA expression is used as a known control for FGR pregnancies [37]. As expected, IL-10 mRNA expression relative to 18S rRNA significantly decreases in expression in placentae from FGR compared with placentae from uncomplicated gestation-matched control pregnancies (*p* < 0.001).

#### Placental NLRP3 mRNA is Increased in FGR Pregnancies

NLRP3 mRNA expression was analysed in the pre-term and term groups, as shown in Figure 3. There is no statistically significant difference in *NLRP3* gene expression between pre-term control and term control, nor between pre-term FGR and term FGR. However, a significant 2-fold increase in NLRP3 mRNA expression is observed in pre-term FGR (*p* < 0.001). Similarly, term FGR samples also show a statistically significant 3.7-fold increase in NLRP3 mRNA expression in comparison to term control samples (*p* < 0.001).

### 3.3. Placental NLRP3 Protein is Increased in FGR Pregnancies

A representative Western blot for placental NLRP3 expression in FGR, and gestation-matched control pregnancies is shown in Figure 4a. An immunoreactive NLRP3 protein at 110 kDa is evident in all placental samples tested. In Figure 4b, semi-quantitative densitometry of the immunoreactive NLRP3 protein normalised to GAPDH demonstrates a significant increase in NLRP3 protein in the placentae obtained from FGR compared with gestation-matched control pregnancies (n = 5 in each group, *p* < 0.05, *t*-test).

### 3.4. NLRP3 Protein is Expressed in the First Trimester and Term Trophoblasts

The spatial distribution of NLRP3 protein in the placentae obtained from first trimester (6–12 weeks), and in term, FGR and term control pregnancies was investigated using immunofluorescence microscopy. As depicted in Figure 5, NLRP3 protein is localised to the villous cytotrophoblasts (VCT), syncytiotrophoblast (STB) and in some stromal cells (STR) in the first trimester villous tissues. As shown in Figure 5, NLRP3 protein is localised to STB, endothelial cells (EC) surrounding the fetal capillaries, and in STR in both FGR and control placentae. A qualitative increase in the intensity of immunoreactive NLRP3 protein is shown in the FGR placentae compared with control.

### 3.5. NLRP3 Protein Expression in an Inflammation-Induced Model of First Trimester Trophoblast Organoid Cultures In Vitro

Human placental organoid cultures were prepared from first trimester placental tissues obtained from termination of pregnancies, collected at 6–12 weeks of gestation. Trophoblast organoids were maintained and sub-cultured to passage 3, and then treated with LPS (1 ng/mL) to simulate inflammation-induced changes in trophoblast growth and NLRP3 expression. Trophoblast growth was assessed using immunoreactive protein expression for the proliferation marker, Ki67. The protein localisation of Ki67 and NLRP3 in LPS-treated organoids, compared to untreated control, was investigated using immunofluorescence microscopy, while the total protein content of NLRP3 and caspase-1 in the culture media collected following LPS treatment was quantified using an immunoassay.

Figure 6a shows representative images of the immunoreactive protein localisation for Ki67 and NLRP3 in placental organoids (n = 4) treated with LPS when compared to untreated (UT) control organoids. The immunoreactive protein Ki67 was used to distinguish the proliferating VCTs from the fusion phenotypes of STBs in both UT and LPS-treated organoids. A qualitative decrease in immunoreactive Ki67 is observed in LPS-treated organoids compared to UT controls. NLRP3 protein localisation is shown in both UT controls, and in LPS treated organoids in both VCT and STB. However, a qualitative increase in the immunoreactive NLRP3 protein is observed in LPS treated organoids compared to UT controls. Quantification of the culture media, following 72 h treatment with LPS, shows significantly increased protein concentrations for NLRP3 and caspase-1 in LPS-treated organoid cultures when compared to UT controls (Figure 6b).

### 3.6. The Effect of LPS on NLRP3 Mediated BeWo Cell Function In Vitro

The effect of LPS stimulation on NLRP3; caspase-1; pro-inflammatory IL-6, IL-1β, IL-18, and anti-inflammatory IL-10 cytokines; trophoblast differentiation marker, CGB/βhCG; and apoptosis was investigated in cultured choriocarcinoma-derived BeWo cells. Confluent cultures treated in the presence of 1 ng/mL LPS show significant increases in NLRP3, caspase-1, IL-6, IL-1β, IL-18, CGB, and caspase-3 and 8 mRNA, and a significantly decreased IL-10 mRNA relative 18S rRNA (Figure 7); increased protein concentrations of NLRP3, caspase-1, IL-6, IL-1β, IL-18, βhCG, and an increased activity of caspase-3 and 8; and decreased IL-10 proteins when compared to untreated cells (Figure 8).

To further assess the role of NLRP3 in LPS-treated BeWo cell function, NLRP3 expression was silenced using short-interference RNA (siRNA). siNLRP3 treatment significantly decreases LPS-induced BeWo cell expression of NLRP3, caspase-1, IL-6, CGB, and caspase-3 and 8 mRNA relative to *18S rRNA*, compared to siCONT treated cells (Figure 9). Similarly, protein concentrations of NLRP3, caspase-1, IL-6, IL-1β, IL-18, βhCG, and caspase-3 and 8 activities significantly decrease in siNLRP3 treated cells compared to siCONT treated cells, while IL-10 protein concentrations significantly increase in siNLRP3+LPS treated BeWo cells, compared to siCONT treated cells (Figure 10).

### 3.7. NLRP3 and Caspase-1 Expression is Increased in the Murine Model of Inflammation In Vivo

Placental NLRP3 and caspase-1 protein expression associated with inflammation was investigated using LPS-induced inflammation in a murine model of pregnancy in vivo. No changes in maternal body weight or fetal resorptions were observed in our model. However, placental weight in LPS-treated mice significantly decreased compared to saline only treated control mice (23.8 ± 2.19 mg vs. 44.0 ± 1.86 mg, n = 6, *p* < 0.05). Fetal weight also significantly decreased in LPS-treated mice compared to saline only treated control mice (563.2 ± 20.9 mg vs. 885.3 ± 55.45 mg, *p* < 0.05).

The spatial distribution and localisation of the candidate inflammasomes, NLRP3 and caspase-1 proteins, were determined in placental tissue sections using immunohistochemistry. As shown in Figure 11, NLRP3 and caspase-1 protein localise in the spongiotrophoblast (Sp), and in the labyrinth (Lab) zones in the control and LPS-treated placentae. Although there are no marked histopathological changes associated with the placentae obtained from LPS-treated mice, an increased intensity in the immunoreactive protein for both NLRP3 and caspase-1 is observed in the Lab zone of the placentae obtained from LPS-treated mice compared to control mice.

## 4. Discussion

This study included a clinically well-defined cohort of FGR-affected pregnancies that represented the severe end of the FGR spectrum, with the fetuses showing reduced growth by the late second, and early third, trimesters [20,21,22]. FGR pregnancies were selected based on the inclusion criteria for a birth weight less than the 10th centile for gestational age using Australian growth charts, and any two of the additional criteria diagnosed on antenatal ultrasound, including abnormal end-diastolic flow in the umbilical artery, oligohydramnios, or asymmetric growth of the fetus [20,21,22]. Analysis of the patient characteristics showed that the newborn birthweight and placental weight significantly decreased in FGR pregnancies compared with that of the gestation-matched controls, as expected. These findings are consistent with previous reports on lower infant birth weight and placental weight in FGR pregnancies [38,39]. Recent studies by Freedman et al. (2019) reported that placental surface area and thickness are associated with lower birth weight [40]. This was also supported by Liu et al. (2021), suggesting that there is an association between reduced placental weight and decreased birth weight, resulting in an increased risk of developing FGR [38]. Taken together, our study is consistent with previous studies in showing that newborn birth weight is directly proportional to placental weight in these clinically well-defined cohort of FGR pregnancies. Our study carefully selected uncomplicated control pregnancies to match the gestation of the FGR samples. Using these clinical samples, previous studies from our laboratory reported consistent gene expression differences for many placental growth control genes [20,21,22,36,41,42,43,44,45], and in a recent study we report an increased expression of NLRP7 inflammasome in the placentae from FGR pregnancies [17].

Here in this study, we focused on identifying the presence and absence of inflammasomes in the placentae from FGR compared with control pregnancies, using a custom designed Fluidigm Biomark^TM^ array. Our previous studies demonstrate gene expression differences for the components of the placental serotonin signalling pathway in FGR pregnancies compared with control pregnancies, using a pathway specific custom designed Fluidigm Biomark^TM^ Array (Fluidigm Corporation, San Francisco, CA, USA), as reported by Ranzil et al. [23]. We analysed 14 genes that were involved in the inflammasome pathway. Further validation using real-time PCR was performed on the seven DAMPs (*NLRP3, NLRC5, CASP1, CASP3, CASP8, NOD2,* and *NFκB1*) and four cytokines (*IFN-γ, IL-1β, IL-6*, and *IL-10*) genes. Out of the 11 genes, the gene expression of 10 relative to *18S rRNA* significantly increased in the placentae from FGR pregnancies compared with control. As expected, the anti-inflammatory cytokine, *IL-10* mRNA expression significantly decreases in the placental tissues of FGR pregnancies compared with the gestation-matched control pregnancies. *TLR* genes were not validated as they belong to the PAMPs, which the control infection-mediated signalling pathways, since the human pregnancies included in this study did not show any signs of infection.

Even though the relationship between gene expression, birth weight, and placental weight is not completely understood, studies report the activation of CASP1 and pro-inflammatory cytokines (IL-1β and tumour necrosis factor (TNF)-α) correlate with reduced placental weight during placental malaria infection [46]. This implies that placental infection-induced inflammation is associated with lower placental weight. As mentioned above, the increase in pro-inflammatory cytokines, and activation of the components of the inflammasomes cascade, is consistently reported in pregnancies associated with placental inflammation [17,37,47,48,49,50,51,52,53,54,55,56,57,58,59]. This study demonstrates that the reduction of placental weight and birth weight in FGR may be associated with sterile inflammation, specifically when there are no signs of infection associated with the FGR placentae included in this study.

This study also demonstrates that placental *NLRP3* expression significantly increases in FGR-affected pregnancies, compared with gestation-matched control pregnancies. As one of the most studied inflammasomes, it is expected that NLRP3 plays a role in a variety of pathologies. NLRP3 is active in immune cells, such as macrophages, during the innate immune response through the activation of CASP1 and IL-1β [52]. However, the activation of NLRP3 is not limited to bacterial exposure, but also to various types of damage-associated molecular patterns (DAMPs), including uric acid, cholesterol crystals, palmitic acid, ROS, and HMGB1 [12,50,55,60,61]. In the placenta, previous studies show the upregulation of *NLRP3* in decidual stromal cells and trophoblasts following LPS stimulation, indicating that NLRP3 may contribute to the placental innate immune response [62]. Furthermore, DAMP-induced NLRP3 levels have been extensively investigated in placental inflammation and dysfunction through the accumulation of cytokines (e.g., IL-1β), all of which were implicated in various pregnancy complications including preeclampsia [55] and spontaneous pre-term labour [63]. In the pathogenesis of preeclampsia, the activation of NLRP3 by DAMPs leads to an increased blood pressure through sympathetic nervous system activation, activation of the renin–angiotensin–aldosterone system (RAAS), tubulointerstitial inflammation, and placental abruption [56]. Furthermore, previous studies by Stødle et al. (2018) and Weel et al. (2017) report an increased expression of *NLRP3*, localised to the syncytiotrophoblast layer in the placentae from women with preeclampsia, compared with normotensive women [53,57]. Taken together, the above studies demonstrate that the overexpression of *NLRP3* is involved in the pathogenesis of preeclampsia. Building on this, our study is the first to demonstrate a similarly increased expression for *NLRP3* in the placentae from FGR-affected pregnancies. Hence, reducing NLRP3 activation may be a therapeutic target for FGR, given that other studies demonstrate the utility of inhibiting NLRP3 activation using β-hydroxybutyrate and the specific inhibitor MCC950, to reduce NLRP3 inflammasome-mediated preterm birth and fetal death in various animal models [35,58].

FGR is characterised by impaired trophoblast fusion and syncytialisation [64,65]. In this study, we report that NLRP3 is localized in the syncytiotrophoblast, endothelial cells, and sporadic stromal cells of both term control and term FGR-affected placentae. The spatial distribution of NLRP3 in control and FGR placentae suggests that the quantitative differences in NLRP3 expression observed between FGR and control placentae reflect changes in NLRP3-expressing cell types in both groups.

Given the close relationship between NLRP3 and CASP1, several studies demonstrate the contribution of CASP1 in various inflammatory processes, especially in pregnancy-related complications. The recruitment of CASP1 is predominantly regulated by the activation of the components of the inflammasome complex, following exposure to DAMPs or microbial components [66,67]. Subsequently, this leads to the maturation of the pro-inflammatory cytokine, IL-1β. These findings are also observed during normal pregnancy, as the production of IL-1β in the umbilical cord, maternal blood, and term placenta depends on CASP1 activation [66]. In our study, we report a significant increase in mRNA expression for *CASP1* in FGR compared to control pregnancies. These findings are supported by studies that show that *CASP1* is detected, and its expression levels increased, in placentae from premature rupture of membrane (PROM) patients [68], women with a high BMI [69], and preeclamptic women [53], all of which are associated with inflammation. Furthermore, Brien et al. (2017) report decreased expression of *IL-6* and *IL-1β* following CASP1 inhibition in monosodium urate crystal-induced inflammation in human term placentae [48]. Taken together, the increased mRNA expression of *CASP1* is likely to be involved in the inflammatory pathways observed in FGR pregnancies.

We also demonstrate an increased *CASP3* mRNA expression in FGR pregnancies, compared with gestation-matched controls. Several studies suggest that CASP3 is responsible for proteolytic activity associated with apoptosis, making it a potential marker of apoptosis [70]. CASP8, a pro-apoptotic protease, generates apoptotic pathways in response to external stimuli [71]. This protease is also essential to the priming of the NLRP3 inflammasome in response to TLRs stimulation by bacterial LPS [72]. Few studies have investigated the role of CASP8 in the pathogenesis of pregnancy-related pathologies, let alone FGR. Previous studies show that an increased expression of *CASP8* during normal pregnancy results in reduced placental growth [73]. Furthermore, Zhao et al. (2009) suggest that an inhibition of CASP8 activity results in a decreased rate of embryonic malformations in murine embryos with diabetic embryopathy, in which there is a suppression of apoptosis in the neural tube epithelium [74]. In our study, *CASP8* mRNA expression is reported to increase in placentae from FGR pregnancies compared with controls, hence, further research is warranted to understand the molecular mechanisms by which CASP8 may contribute to the pathogenesis of FGR.

We determined the levels of pro-inflammatory cytokine IL-6, which is one of the signalling molecules involved in the maternal immune response. Aggarwal et al. (2019) and Ribeiro et al. (2022) report that levels of IL-6 significantly increase in placental tissues and maternal serum samples in women with preeclampsia [75,76], though more research is needed to determine the extent to which IL-6 contributes to disease progression. In a study on maternal peripheral blood lymphocytes, IL-6 levels increased in FGR pregnancies with placental insufficiency, and IL-6 acts as a potential marker of the inflammatory process [37]. Taken together, the upregulation of *IL-6* may be associated with the pathogenesis of placental dysfunction in FGR pregnancies.

This study reports a significant decrease in anti-inflammatory cytokine *IL-10* mRNA expression in the placentae from FGR pregnancies compared with gestation-matched controls. Rivera et al. (1998) [77] propose that IL-10 is a potential therapeutic for perinatal complications due to its aberrant expression in various pregnancy-related pathologies. Aggarwal et al. (2019) [75] and Al-Azemi et al. (2017) [37] report on the decreased levels of IL-10 in both placental tissues and serum in preeclamptic women, and maternal blood lymphocytes in FGR cases with placental insufficiency, respectively. The lower levels of IL-10 are shown to contribute to an imbalance in the pro- and anti-inflammatory cytokine concentrations, which subsequently exacerbate inflammatory responses in both pathologies [37]. In our study, the decreased *IL-10* mRNA expression in idiopathic FGR placentae is consistent with previous studies [37,75], suggesting a role for *IL-10* in the pathogenesis of placental dysfunction in FGR pregnancies.

Using the human trophoblast organoids in vitro model system, this study is the first to demonstrate that an increased immunoreactivity for NLRP3 protein is associated with an inflammation-induced reduced trophoblast growth in LPS-induced organoids, as indicated by a reduction in the immunoreactive Ki67 protein. This observation is further validated by the concentrations of NLRP3, and caspase-1 released in the culture media. With further analysis of the functional consequences of LPS-induced inflammation in cultured BeWo cells, our study demonstrates that increased NLRP3 and caspase-1 expression/concentrations are associated with increases in production of βhCG, IL-1β, IL-18, and IL-6, and activity of caspase-3 and caspase-8, suggesting that NLRP3 may have a direct or indirect regulation on trophoblast differentiation. Expression of specific molecular markers and the production of hCG and progesterone are some of the functional consequences of the process of trophoblast differentiation or syncytialisation [36]. In this study we observed that *NLRP3* siRNA inactivation in LPS-treated BeWo cells significantly reduces *CGB* mRNA, suggesting that *NLRP3* has a negative regulatory role in *CGB* transcription. In agreement to the above observation, in vitro differentiation of VCT from FGR-affected pregnancies are shown to have significantly higher levels of syncytialisation and hormone secretion, compared to trophoblasts derived from uncomplicated pregnancies [65]. However, the underlying mechanisms of how *NLRP3* is involved in the process of trophoblast differentiation or syncytialisation remain elusive.

By using an inflammation-induced murine model of pregnancy in vivo, this study further demonstrates an association with increased NLRP3 and caspase-1 expression in the placentae from LPS-induced mice compared to saline only treated controls. Although there were no distinct observable placental histopathological features in the LPS-induced model, placental and fetal weights significantly reduced in the LPS-treated mice, with an associated increase in the protein expression for NLRP3 and caspase-1 in the placentae from LPS-treated mice. However, the functional consequences of increased placental NLRP3 and caspase-1 in the murine model of inflammation-induced FGR warrants further investigation.

## 5. Conclusions

Our study identifies a consistent increased expression of NLRP3 and caspase-1 in the placentae from a clinically well-defined cohort of human FGR, and in both in vitro and in vivo inflammation-induced models of FGR. In summary, our study supports the association between elevated levels of NLRP3 and aberrant inflammation-induced trophoblast dysfunction in FGR. These findings highlight the importance of better understanding the molecular mechanisms involved in placental inflammation-induced FGR and trophoblast function, and in developing potential therapeutic targets to improve feto-placental growth in FGR.

## Figures and Tables

**Figure 1 cells-11-01413-f001:**
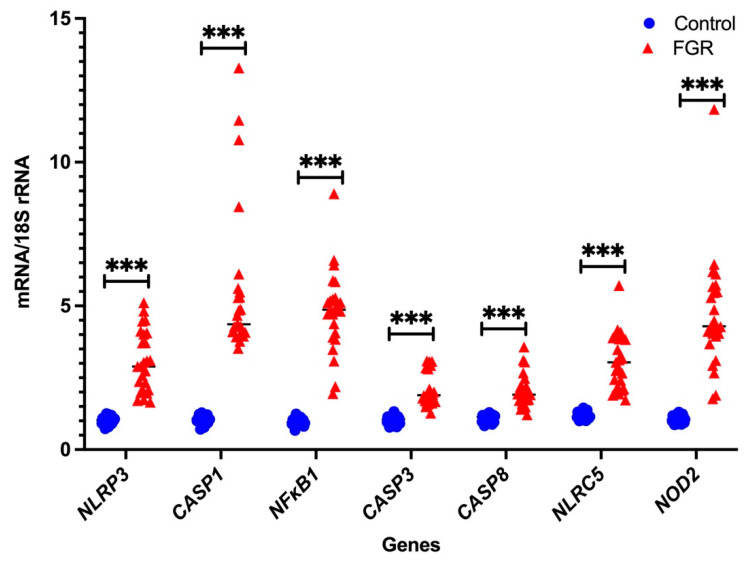
Real-time PCR validation of DAMPs: independent validation of DAMPs-involved mRNA gene expression relative to *18S rRNA*. Statistical significance was determined using a Mann–Whitney U test and indicated by *** (*p* < 0.001).

**Figure 2 cells-11-01413-f002:**
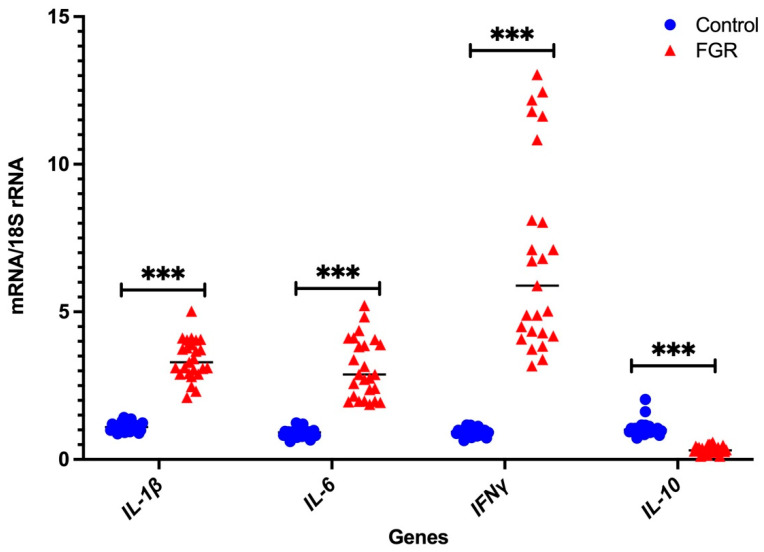
Real-time PCR validation of pro- and anti-inflammatory cytokines: independent validation of mRNA expression of the pro- and anti-inflammatory cytokines relative to *18S rRNA*. Statistical significance was determined using a Mann–Whitney U test and indicated by *** (*p* < 0.001).

**Figure 3 cells-11-01413-f003:**
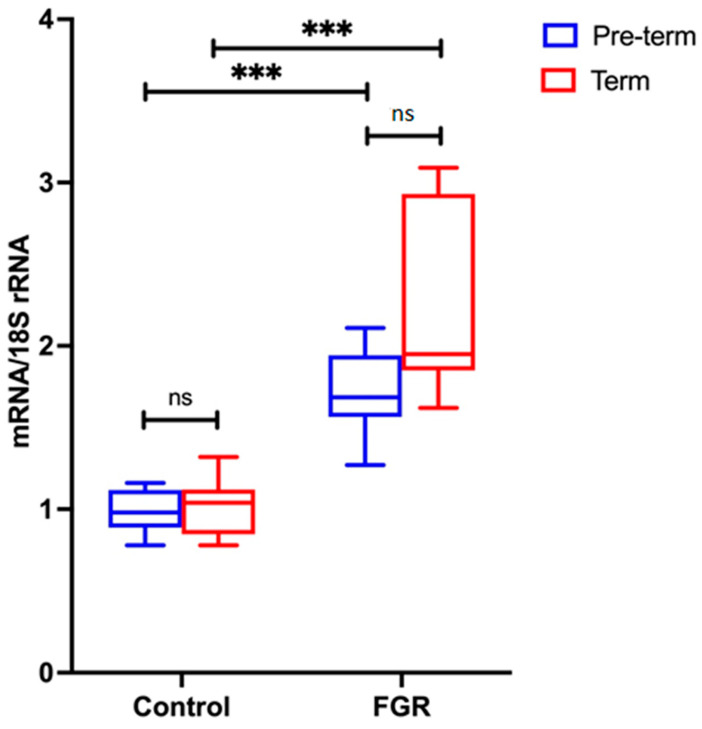
NLRP3 mRNA expression in pre-term and term groups. Data from the pre-term group are shown in blue; from the term group in red. A Mann–Whitney U test was performed to determine statistical differences between the groups. Statistical significance is denoted by *** *p* < 0.001 vs. corresponding group indicated. Non-statistical significance is denoted by ns.

**Figure 4 cells-11-01413-f004:**
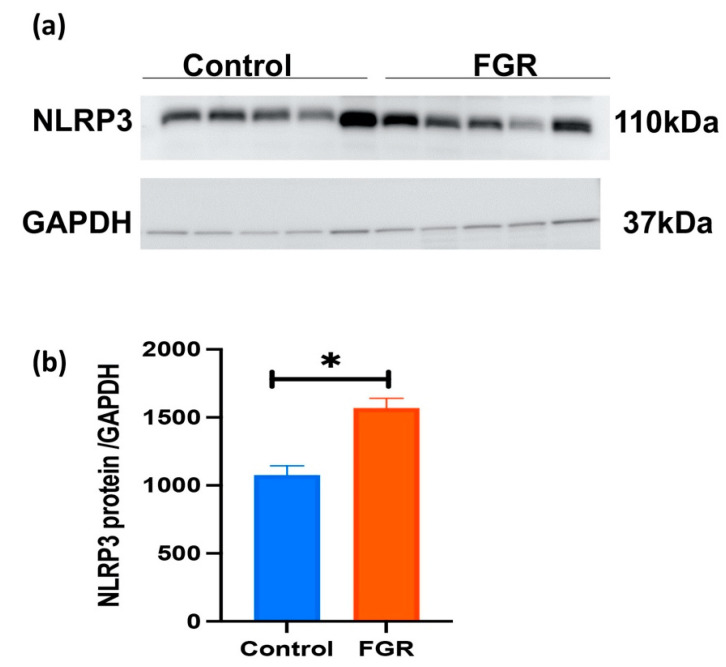
Placental NLRP3 protein expression in control and FGR pregnancies. (**a**) A representative immunoblot shows immunoreactive NLRP3 protein at 110 kDa in placentae obtained from FGR and gestation-matched control pregnancies (n = 5 in each group). GAPDH (37 kDa) was used as a loading control. (**b**). Semi-quantitation of immunoreactive NLRP3 protein normalised to GAPDH demonstrates a significant increase in NLRP3 protein in the placentae from FGR-affected pregnancies compared with gestation-matched control pregnancies. Statistical difference in placental NLRP3 protein content between FGR and control is denoted by * *p* < 0.05 (*t*-test).

**Figure 5 cells-11-01413-f005:**
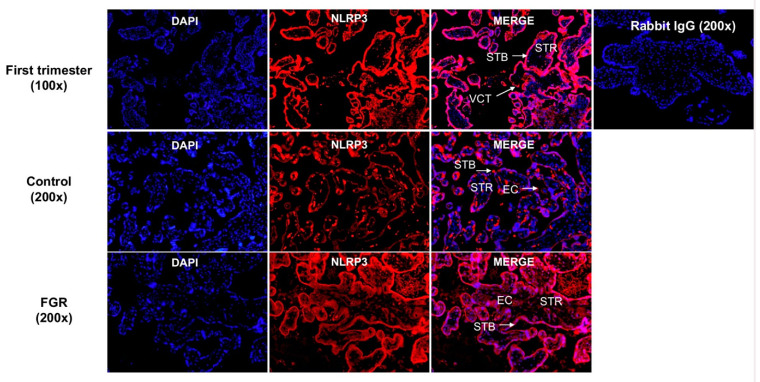
Spatial distribution of NLRP3 in first and third trimester placental tissues. NLRP3 protein localization in first trimester placental tissues (7 weeks of gestation) shows the presence of immunoreactive NLRP3 protein in the villous cytotrophoblasts (VCT), syncytiotrophoblast (STB), and in some stromal cells (STR). Image represents a 100× magnification. Representative images of term control and term FGR placental tissues for NLRP3 immunostaining in STB, STR, and in endothelial cells (EC) surrounding the fetal capillaries in both control and FGR placentae. Image represents a 200× magnification. Rabbit IgG was used as a negative control. Scale bar represents 100 μm.

**Figure 6 cells-11-01413-f006:**
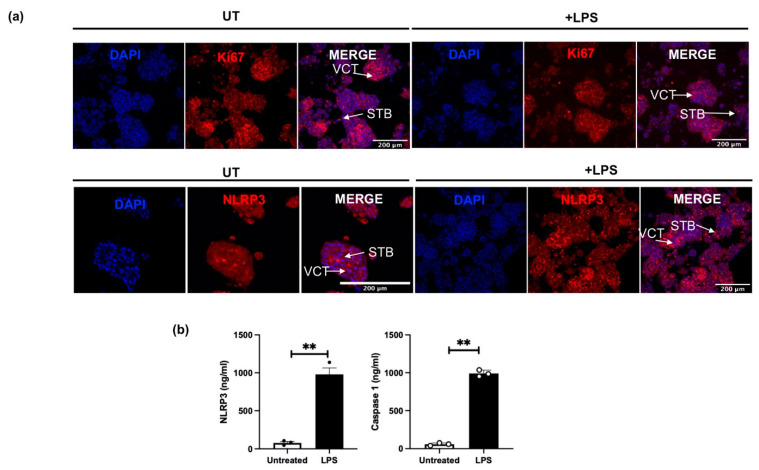
Expression of Ki67 and NLRP3 in human trophoblast organoids obtained from first trimester pregnancies (6–12 weeks of gestation). (**a**) Ki67, a marker for proliferating cells in the VCT, was detected in red in both untreated (UT) and in LPS-treated trophoblast organoids. NLRP3 protein localisation in the VCT and STB is shown in red in both UT and LPS-treated trophoblast organoids. Nuclear staining was performed using DAPI. (**b**) shows concentrations of NLRP3 and caspase-1 in the human placental organoids cultured over 72 h following treatment with or without LPS. Protein concentration of NLRP3 and caspase-1 significantly increased in the conditioned media obtained from LPS treated placental organoids, compared with untreated organoid cultures (n = 3). Statistical difference in NLRP3 and caspase-1 protein between untreated and LPS treated organoid cultures is denoted by ** *p* < 0.005 (*t*-test).

**Figure 7 cells-11-01413-f007:**
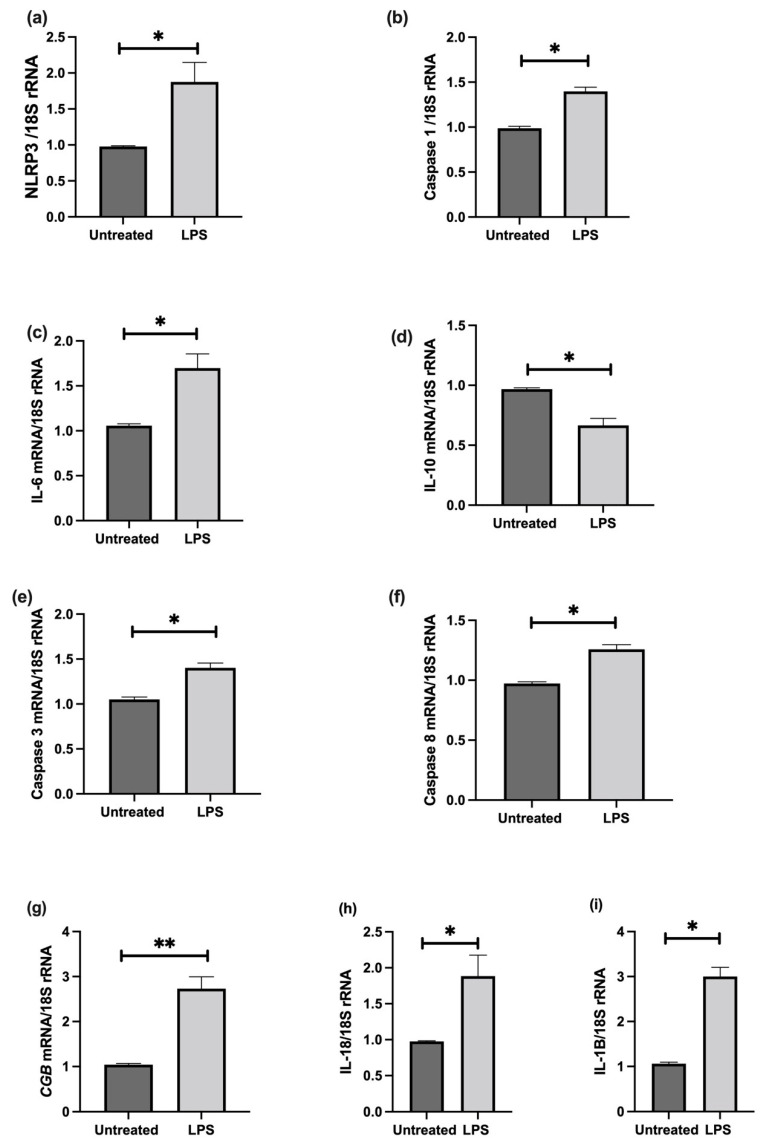
Effect of LPS on BeWo cell function in vitro. Confluent cultures treated in the presence of 1 ng/mL LPS show significant increases in (**a**) NLRP3, (**b**) caspase-1, (**c**) IL-6, (**i**) IL-1β, (**h**) IL-18, (**g**) CGB, and (**e**) caspase-3 and (**f**) Caspase-8 mRNA, and a significantly decreased (**d**) IL-10 mRNA relative *18S rRNA* (**a**–**i**). Significant differences are denoted by * *p* < 0.05; ** *p* < 0.005.

**Figure 8 cells-11-01413-f008:**
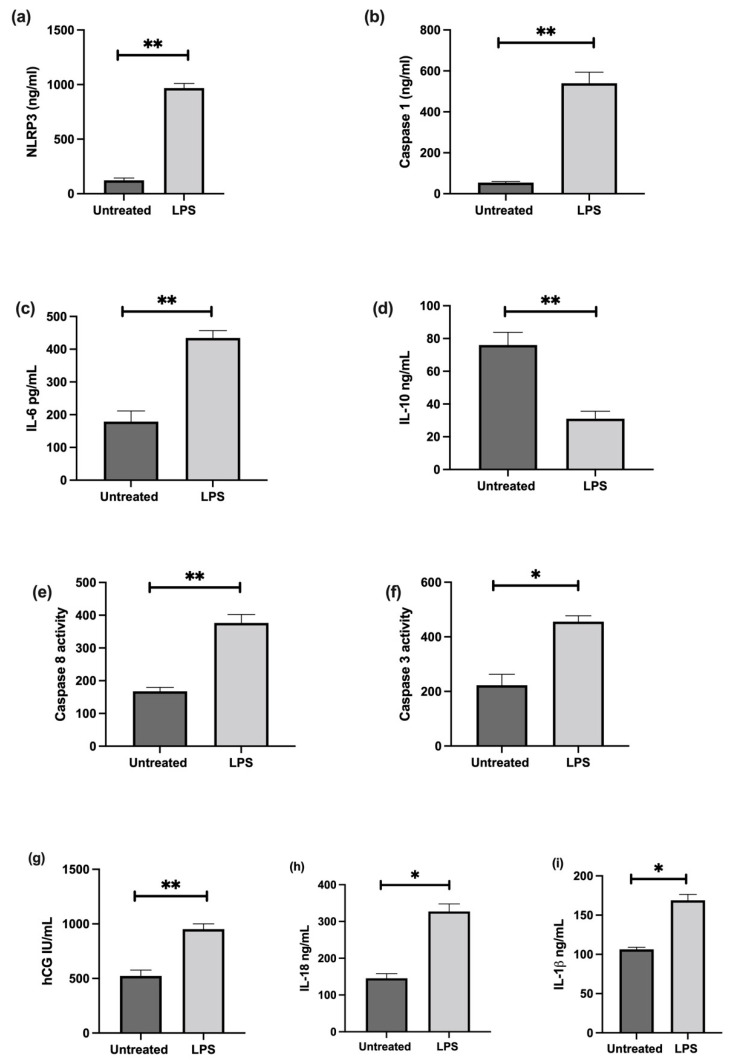
Effect of LPS on protein expression of NLRP3, caspase-1, pro- and anti-inflammatory cytokines, and trophoblast differentiation marker; and activity of apoptosis markers in BeWo cells. LPS treatment increases protein concentrations of (**a**) NLRP3, (**b**) caspase-1, (**c**) IL-6, (**i**) IL-1β, (**h**) IL-18, and (**g**) βhCG, and increases activity of (**f**) caspase-3 and (**e**) 8; but decreases (**d**) IL-10 protein expression when compared to untreated cells. Significant differences are denoted by * *p* < 0.05; ***p* < 0.005.

**Figure 9 cells-11-01413-f009:**
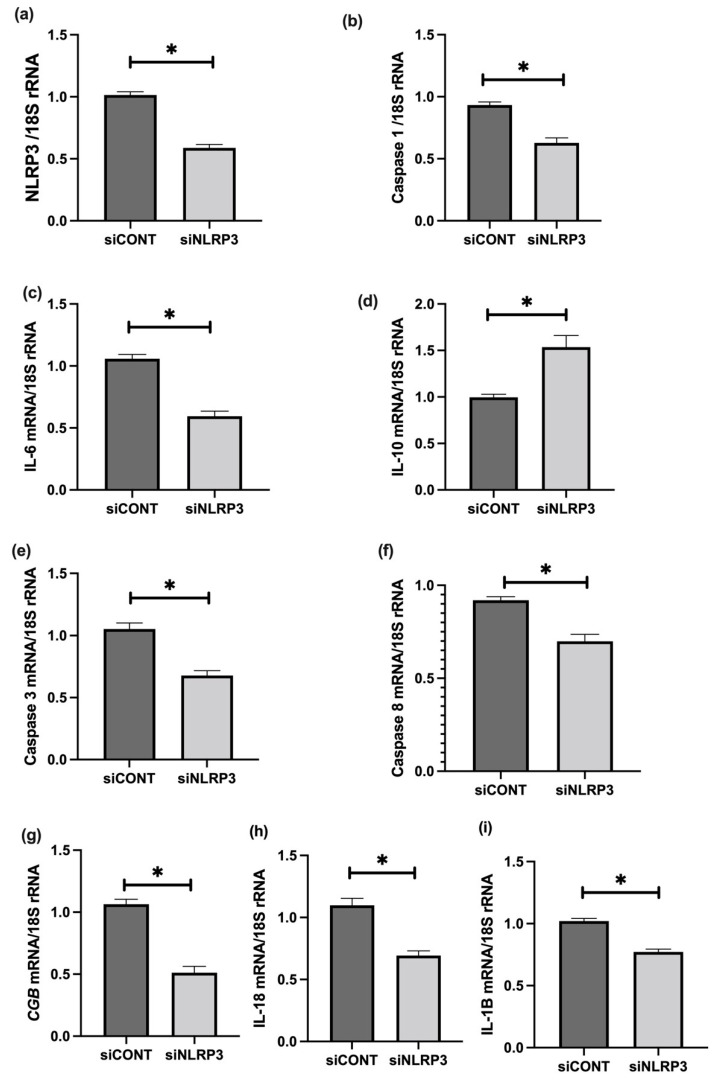
Effect of NLRP3 siRNA inactivation and LPS treatment on mRNA expression of NLRP3, caspase-1, pro- and anti-inflammatory cytokines, trophoblast differentiation, and apoptosis markers in BeWo cells. siNLRP3 treatment significantly decreases LPS-induced BeWo cell expression of (**a**) NLRP3, (**b**) caspase-1, (**c**) IL-6, (**i**) IL-1β, (**h**) IL-18, (**g**) CGB, and (**e**) caspase-3 and (**f**) Caspase-8 mRNA, but increases (**d**) IL-10 mRNA relative to *18S rRNA*, compared to siCONT treated cells. Significant differences are denoted by * *p* < 0.05.

**Figure 10 cells-11-01413-f010:**
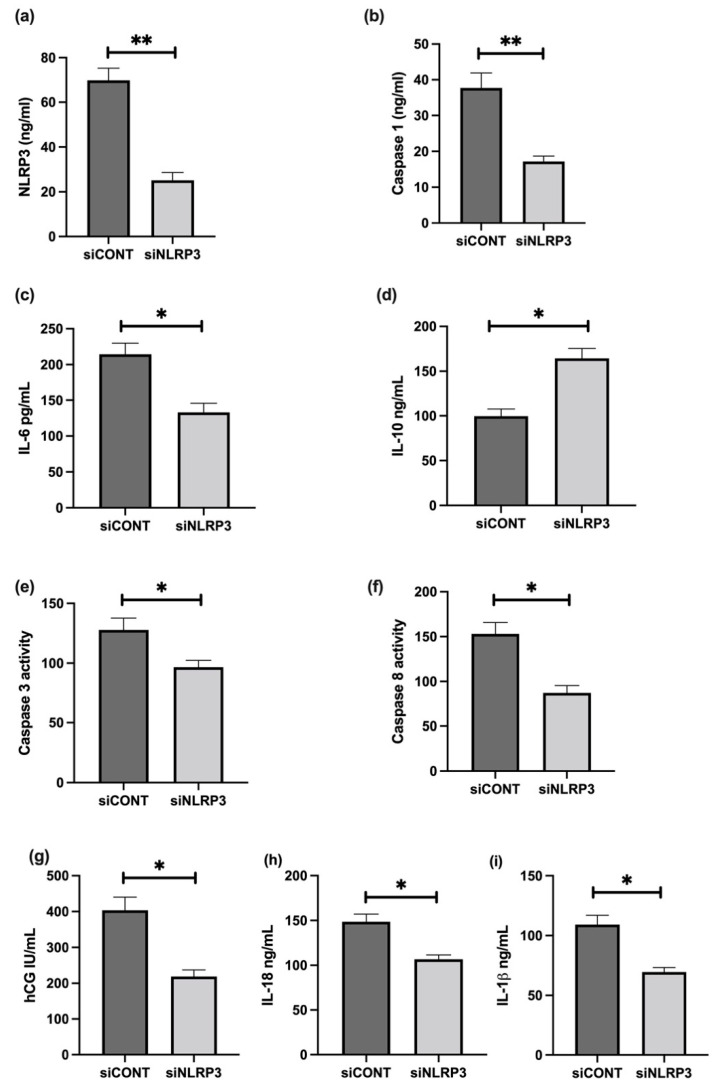
Effect of NLRP3 siRNA inactivation and LPS on protein expression of NLRP3, caspase-1, pro- and anti-inflammatory cytokines, trophoblast differentiation marker, and activity of apoptosis markers in BeWo cells. Protein concentrations of (**a**) NLRP3, (**b**) caspase-1, (**c**) IL-6, (**i**) IL-1β, (**h**) IL-18, (**g**) βhCG, and (**e**) caspase-3 and (**f**) caspase- 8 activities significantly decrease in siNLRP3 treated cells compared to siCONT treated cells. In contrast, (**d**) IL-10 protein concentrations significantly increase in siNLRP3 + LPS treated BeWo cells, compared to siCONT treated cells. Significant differences are denoted by * *p* < 0.05; ** *p* < 0.005.

**Figure 11 cells-11-01413-f011:**
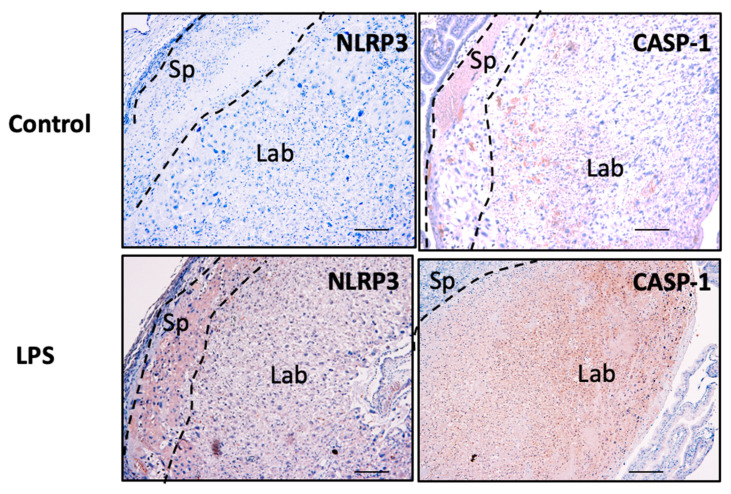
Placental NLRP3 and caspase-1 protein in murine pregnancy in vivo. Placental NLRP3 and caspase-1 expression levels associated with inflammation are investigated using immunohistochemistry in the placentae obtained from LPS-induced inflammation in a murine model of pregnancy. As shown, NLRP3 localisation is observed in the spongiotrophoblast (Sp) and labyrinth (Lab) in the placentae from both control and LPS-treated mice. Scale bar denotes 100 μm.

**Table 1 cells-11-01413-t001:** Patient characteristics of third trimester FGR and uncomplicated pregnancies.

Characteristics	Control (n = 25)	FGR (n = 25)	Significance
Gestational age (weeks)	34.44 ± 3.959	36.12 ± 3.232	*p* = 0.107
Maternal age (years)	34 ± 5.323	31.2 ± 5.008	*p* = 0.061
Parity		*p* = 0.089	
Primaparous	9 (36%)	16 (64%)	
Multiparous	16 (64%)	9 (36%)	
Mode of delivery		*p* = 0.300	
Vaginal delivery	6 (24%)	9 (36%)	
Caesarean in labour	1 (4%)	3 (12%)	
Caesarean not in labour	18 (72%)	13 (52%)	
Newborn Characteristics			
Gender		*p* ≥ 0.999	
Male	11 (44%)	12 (48%)	
Female	14 (56%)	13 (52%)	
Placental weight (g)	506 ± 144.1	395.9 ± 124	*p* = 0.006
Birth weight (g)	2474 ± 875.9	1968 ± 662.9	*p* = 0.026
Birth weight percentile			
10th–90th	25 (100%)	0	
5th–10th		12/25 (48%)	
3th–5th		11/25 (44%)	
<3th		2/25 (8%)	

**Table 2 cells-11-01413-t002:** Clinical criteria of FGR samples in this study.

Clinical Characteristics	Number of Samples (%)
BW < 10th percentile	25/25 (100%)
Abnormal umbilical artery Doppler velocimetry	
Elevated	5/25 (20%)
Reversed	6/25 (24%)
Absent	8/25 (32%)
Normal	4/25 (16%)
Not recorded	2/25 (8%)
Asymmetric growth	
HC:AC ratio > 1.2	25/25 (100%)
Amniotic fluid index (AFI)	
Normal (AFI = 7)	7/25 (28%)
Polyhydramnios (AFI > 7)	7/25 (28%)
Oligohydramnios (AFI < 7)	11/25 (44%)

HC, head circumference; AC, abdominal circumference.

**Table 3 cells-11-01413-t003:** Fluidigm Biomark^TM^ Array showed the presence of DAMPs.

Genes	Control (n = 25)	FGR (n = 25)
*NLRP3*	1.213 ± 0.851	1.099 ± 1.204
*CASP1*	1.001 ± 1.305	1.552 ± 10.474
*NFκB1*	1.303 ± 0.905	0.937 ± 1.189
*CASP3*	1.384 ± 1.423	0.529 ± 0.845
*CASP8*	1.256 ± 0.982	0.549 ± 0.397
*NLRC5*	1.197 ± 1.427	1.508 ± 2.138
*NOD2*	1.259 ± 1.117	1.767 ± 6.981

**Table 4 cells-11-01413-t004:** Fluidigm Biomark^TM^ Array showed the presence of PAMPs.

Genes	Control (n = 25)	FGR (n = 25)
*TLR2*	0.958 ± 2.138	1.093 ± 4.707
*TLR5*	0.685 ± 1.180	0.397 ± 4.318
*TLR6*	0.620 ± 1.469	0.135 ± 0.658

**Table 5 cells-11-01413-t005:** Fluidigm Biomark^TM^ Array showed the presence of cytokines.

Genes	Control (n = 25)	FGR (n = 25)
*IL-1β*	0.822 ± 1.208	2.724 ± 15.483
*IL-6*	1.204 ± 0.579	0.900 ± 4.033
*IFNγ*	0.958 ± 1.596	11.7 ± 119.944
*IL-10*	0.969 ± 0.950	1.163 ± 28.943

## Data Availability

The datasets analysed or generated during the study are stored within the Monash University data storage.

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
