# Peer review of "The Placental NLRP3 Inflammasome and Its Downstream Targets, Caspase-1 and Interleukin-6, Are Increased in Human Fetal Growth Restriction: Implications for Aberrant Inflammation-Induced Trophoblast Dysfunction"

_cells, 2022, doi:10.3390/cells11091413_

Round 1

Reviewer 1 Report

This study investigates the relationship between NLRP3 inflammasomes and the pathogenesis of fetal growth restriction. In this study, rt-PCR, Fluidigm BiomarkTM Array, immunohistochemistry, Immunofluorescence, Western immunoblotting et al were used to explore the results. They finally concluded that NLRP3 inflammasome in inflammation-induced aberrant trophoblast function, which may contribute to FGR. The writing is fine. The author should answer the following questions:

  1. In Table 3, the expression profiles of NLRP3, NFkB1, CASP3, and CASP8 in the control group were higher than those in the FGR group, while the results in Figure 1 were reversed.
  2. In Table 5, the expression of IL-6 in the control group was higher than that in the FGR group, and the expression of IL-10 in the FGR group was higher than that in the control group. The results of these two terms in Figure 2 are also opposite.

Author Response

Response to Reviewers’ Comments:

The authors thank the reviewers for their insightful and constructive comments. We have endeavoured to address all concerns raised by the reviewers and believe that the revised manuscript is suitable for consideration to publication in Cells.

Reviewer 1

Comments and Suggestions for Authors

This study investigates the relationship between NLRP3 inflammasomes and the pathogenesis of fetal growth restriction. In this study, RT-PCR, Fluidigm BiomarkTM Array, immunohistochemistry, Immunofluorescence, Western immunoblotting et al were used to explore the results. They finally concluded that NLRP3 inflammasome in inflammation-induced aberrant trophoblast function, which may contribute to FGR. The writing is fine. The author should answer the following questions:

  1. In Table 3, the expression profiles of NLRP3, NFkB1, CASP3, and CASP8 in the control group were higher than those in the FGR group, while the results in Figure 1 were reversed.

Authors’ Response: The custom-designed Fluidigm Biomark array used in this study was mainly used to screen for the presence or absence of components of the inflammasomes in the FGR and control placental tissues.  Gene amplification was observed for all genes tested in both FGR and control samples, however, there was no statistical difference in the gene expression analyses for all the genes tested between FGR and the control groups. The primers provided in the array specifically for the house-keeping control genes, GAPDH and 18S rRNA were not pre-validated and showed large variation in their cycle threshold and the relative gene expression for the inflammasome genes, as shown in Table 3. In the revised manuscript, we have now added statements describing the use of the Fluidgim array as a screening tool in lines 125; 338-340; 351 and 355. Further validation for relative quantitation was necessitated using inventoried, pre-validated TaqMan probes for all the genes tested as shown in Figure 1.

  1. In Table 5, the expression of IL-6 in the control group was higher than that in the FGR group, and the expression of IL-10 in the FGR group was higher than that in the control group. The results of these two terms in Figure 2 are also opposite.

Authors’ Response: As described above in response to question 1, pro-inflammatory cytokines including IL-6 and anti-inflammatory IL-10 expression presence and absence was screened using the Fluidigm array as detailed in response 1 above. Further validation using independent real-time PCR showed significant difference in IL-6 (increased) and IL-10 (decreased) expression in FGR compared to control.

Reviewer 2 Report

Nice tailored artice

Author Response

We thank the reviewer for their constructive comments.

Reviewer 3 Report

1-Authors need to review a results section.
The results can be more precise and objective. There is a lot of repeated information about the methodology that has already been described in this section.

2-In the discussion section, I suggest that the authors add this reference to interleukin IL-6, which is found to be increased in preeclampsia.

-Ribeiro VR, Romao-Veiga M, Nunes PR, de Oliveira LRC, Romagnoli GG, Peracoli JC, Peracoli MTS. Immunomodulatory effect of vitamin D on the STATs and transcription factors of CD4+ T cell subsets in pregnant women with preeclampsia. Clin Immunol. 2022 Jan;234:108917. doi: 10.1016/j.clim.2021.108917. Epub 2021 Dec 29. PMID: 34973430.

3-The acronym NLRP3 is described in two different ways. Adopt the NLRP3 standard throughout the text.

Author Response

Reviewer 3

Comments and Suggestions for Authors

1-Authors need to review a results section.
The results can be more precise and objective. There is a lot of repeated information about the methodology that has already been described in this section.

Authors’ Response: In the revised manuscript, we have removed repetitiveness and concisely described the results section as suggested by this reviewer.

2-In the discussion section, I suggest that the authors add this reference to interleukin IL-6, which is found to be increased in preeclampsia.

-Ribeiro VR, Romao-Veiga M, Nunes PR, de Oliveira LRC, Romagnoli GG, Peracoli JC, Peracoli MTS. Immunomodulatory effect of vitamin D on the STATs and transcription factors of CD4+ T cell subsets in pregnant women with preeclampsia. Clin Immunol. 2022 Jan;234:108917. doi: 10.1016/j.clim.2021.108917. Epub 2021 Dec 29. PMID: 34973430.

Authors’ Response: In the revised manuscript, in lines 684-686, we have added the suggested reference.

3-The acronym NLRP3 is described in two different ways. Adopt the NLRP3 standard throughout the text.

Authors’ Response: The acronym for the mouse gene is expressed as Nlrp3. Since we investigated the protein localisation in the mouse placental sections, we have adopted NLRP3 (all caps) as suggested by this reviewer. 

Reviewer 4 Report

The authors tried to show that the role of the NLRP3 inflammasomes on fetal growth restriction analyzing human placenta and animal model. Most of the data in this manuscript are convincing and presented by well-designed experiments. Addressing these following concerns would strengthen the conclusions of the manuscript.

Major points:
1. At first, the authors analyzed gene expression using Fluidigm Biomark array. However, there were no differences of all examined gene expression including NLRP3 inflammasome. Contrary, the authors evaluated same gene expression using real-time PCR and all examined gene expression were significantly higher in FGR samples. The authors need to explain why such a big difference is made.
This reviewer think it is necessary to reconsider these points, whether there was a problem with the analysis method using Fluidigm Biomark array, or whether it is really necessary to use analysis data using Fluidigm Biomark array.

2. In discussion, the authors described that FGR may be associated with sterile inflammation specifically when there are no signs of infection associated with the FGR placentae. However, the authors used LPS to examine the role of NLRP3 inflammasome in vivo and in vitro experiments. Is the evaluation using LPS really valid in this study?

3. Since the focus of this research is the NLRP3 inflammasome, it is necessary to measure IL-1b and IL-18 in Figure 6, Figure 7, Figure 8.

Minor points:
1. Methods: Line 157. Please describe the reason for the LPS administration concentration.

2. Methods: Line 248. Please describe the concentration of antibodies in western blotting.

3. Methods: Line 257. Were extracellular NLRP3 and Caspase-1 measured by ELISA? If so, please include information such as the catalog number.

4. Methods: There were no information about siRNA experiment.

5. Figure 5: Since the authors stated that fluorescent immunostaining was also performed on multiple samples with n = 6, is it possible to quantify?

6. Figure 9 and 10: Was there any effect of NLRP3 siRNA alone without LPS treatment?

7. Figure 11: Add the information about maternal body weight change and fetal absorption rate.

8. Discussion and references: The following papers have already been published about the role of NLRP3 inflammasome in inflammation and FGR, so please cite and discuss them. PMID: 32056551 PMID: 34628106 PMID: 30590393 PMID: 30596885 PMID: 32101829

Author Response

Reviewer 4

Comments and Suggestions for Authors

The authors tried to show that the role of the NLRP3 inflammasomes on fetal growth restriction analyzing human placenta and animal model. Most of the data in this manuscript are convincing and presented by well-designed experiments. Addressing these following concerns would strengthen the conclusions of the manuscript.

Major points:
1. At first, the authors analyzed gene expression using Fluidigm Biomark array. However, there were no differences of all examined gene expression including NLRP3 inflammasome. Contrary, the authors evaluated same gene expression using real-time PCR and all examined gene expression were significantly higher in FGR samples. The authors need to explain why such a big difference is made.

This reviewer think it is necessary to reconsider these points, whether there was a problem with the analysis method using Fluidigm Biomark array, or whether it is really necessary to use analysis data using Fluidigm Biomark array.

Authors’ Response: We acknowledge this reviewer’s concern on the presentation of the Fluidigm analysis. As detailed in response to reviewer #1 comment, the Fluidigm Biomark array was used as a screening tool to detect the presence or absence of inflammasome-related genes in the placentae obtained from control vs FGR groups. The array data was analysed using two independent house-keeping genes as recommended by the manufacturer. Gene amplification was observed for all genes tested in both FGR and control samples, however, there was no statistical difference in the gene expression analyses for all the genes tested between FGR and the control groups. Therefore, further validation was carried out using inventoried TaqMan probes was performed using real-time PCR and the relative quantitation was performed using 18S rRNA and the statistical differences between the two groups were determined.

2. In discussion, the authors described that FGR may be associated with sterile inflammation specifically when there are no signs of infection associated with the FGR placentae. However, the authors used LPS to examine the role of NLRP3 inflammasome in vivo and in vitro experiments. Is the evaluation using LPS really valid in this study?

Authors’ response: In the in vitro experiments, we used low-dose LPS (1ng/mL) to simulate low-grade infection as previously described by Baker et al., (2021). This information is now added to the methodology section in line 203.  

As a proof of concept, low-dose LPS is commonly used for studying inflammation-induced changes in the feto-placental gene expression. For example, LPS induced model of inflammation produced trophoblast specific changes in animal models and simulated FGR like phenotypes as previously described by Cotechini et al (2014). More recent studies by Hirata et al. (2021) demonstrated that LPS induced inflammation in murine model of pregnancy significantly enhanced the expression of placental NLRP3 and its down-stream targets.

3. Since the focus of this research is the NLRP3 inflammasome, it is necessary to measure IL-1b and IL-18 in Figure 6, Figure 7, Figure 8.

Authors’ Response: As suggested by this reviewer, we have added quantification of mRNA and the protein concentrations of IL-1B/IL-1b and IL-18 to Figures 7-10.

Minor points:

1. Methods: Line 157. Please describe the reason for the LPS administration concentration.

Authors’ Response: In lines 162-167, 228-230, we have now described the reasons for using low dose LPS to mimic sub-clinical infection and placental inflammation induced changes in the expression of inflammasomes.

2. Methods: Line 248. Please describe the concentration of antibodies in western blotting.

Authors’ Response: In the revised manuscript, in line 274 we have now provided the concentration and dilution of the antibodies.

3. Methods: Line 257. Were extracellular NLRP3 and Caspase-1 measured by ELISA? If so, please include information such as the catalog number.

Authors’ Response: In the revised manuscript, in line 283-288, we have now provided the catalogue numbers and the dilutions of the antibodies.

4. Methods: There were no information about siRNA experiment.

Authors’ Response: We thank the reviewer for pointing out this missed information in the methods section. In the revised manuscript, in lines 203-214, siRNA experimental details are provided.

5. Figure 5: Since the authors stated that fluorescent immunostaining was also performed on multiple samples with n = 6, is it possible to quantify?

Authors’ Response: We believe immunofluorescence provides qualitative information and is mainly used for spatio-temporal localisation and distribution of proteins of interest. Therefore, no quantitation was conducted. 

6. Figure 9 and 10: Was there any effect of NLRP3 siRNA alone without LPS treatment?

Authors’ Response: Pilot data showed no significant change with siNLRP3 treatment alone for cytokine and hCG concentrations, therefore low dose LPS was used to induce inflammation in BeWos.

7. Figure 11: Add the information about maternal body weight change and fetal absorption rate.

Authors’ Response: Maternal body weight did not change and we did not see fetal absorption. This information is added to the revised manuscript text in line 536.

8. Discussion and references: The following papers have already been published about the role of NLRP3 inflammasome in inflammation and FGR, so please cite and discuss them.

Authors’ Response: We have added all additional papers suggested by this reviewer in the discussion section and cited them appropriately.

Round 2

Reviewer 1 Report

The author has answered the question clearly.

Author Response

(The authors gave the same response as above.)

Reviewer 4 Report

The authors have made substantial efforts and included new data to answer questions and comments made by this reviewer. However, some technical issues persist.

In the first submitted manuscript, the authors used 1 µg/ml dose of LPS in vitro study. However, in the revised manuscript, the authors changed the dose of LPS (low-dose 1 ng/ml) in the Material and Methods section. In addition, in Line 505, the authors used 1 µg/ml dose of LPS in the revised manuscript. If the authors changed the LPS concentration and restarted the revised experiment, it is strange that the figures are the same.

Author Response

We thank the reviewer for highlighting the errors in the concentration of LPS in the manuscript text. LPS concentration for all in vitro work were carried out at 1ng/mL, while for the in vivo studies 1 µg/bodyweight was used to study inflammation-induced changes in the components of inflammasomes and their mediators.

In the revised manuscript text,  in lines 442, 478, 491 we have now corrected LPS concentrations to 1ng/mL for the in vitro studies.